# Quantitative Measurement of Radio Frequency Interference for SMOS Mission

Ming Xu , Hongping Li *, Haihua Chen and Xiaobin Yin

College of Marine Technology, Ocean University of China, Qingdao 266100, China;
xuming@stu.ouc.edu.cn (M.X.); chh7791@ouc.edu.cn (H.C.); yinxiaobin@ouc.edu.cn (X.Y.)
* Correspondence: lhp@ouc.edu.cn

**Abstract:** The Earth Exploration Satellite Service (EESS) for passive sensing has a primary frequency allocation in the 1400–1427 MHz band. All emissions unauthorized in this band are called RFI (Radio Frequency Interference). The SMOS (Soil Moisture and Ocean Salinity) mission is greatly perturbed by RFI impeding ocean salinity retrieval, especially in coastal areas such as the SCS (South China Sea), where the observed data has been discarded massively. At present, there is no way to eliminate the RFI impact on the retrieved salinity, other than by detecting and shutting down the emissions from the sources. However, it may be effective in a scientific sense if RFI can be quantified and applied to the salinity retrieval process. Therefore, this study proposes an RFI measuring method that can investigate contamination in both prominent and moderate respects, aroused either by on-site emissions or nearby continents. Based on the proposed method, two levels of hierarchical RFI maps of the SCS region, including the separated one and the merged one, are presented and discussed, indicating more severe contamination in northern and western SCS. Moreover, to verify the generalization of the method on open oceans far from continents, an area in the middle central Pacific is selected and tested. The result shows few or no RFI in this unattended region, which is consistent with the authors' knowledge. This study presents the concept of the "RFI map" to describe the contamination, which will hopefully help researchers comprehend the RFI state intuitively and assist in ocean salinity retrieval statistically.

**Keywords:** radio frequency interference (RFI); measurement; soil moisture and ocean salinity (SMOS); prominent contamination; moderate contamination

## 1. Introduction

SSS (Sea Surface Salinity) plays an important role in the stability of the upper ocean by affecting the thermohaline structure and dynamical processes, indirectly influencing the air–sea interaction and climate events [1]. In addition, salinity intrusion is one of the major concerns in coastal areas that threatens to worsen in changing climatic conditions [2]. Although optical data such as MODIS can derive SSS through multi-band fitting [3], passive microwaves are the only direct means to remotely sense SSS based on physical mechanisms [4].

The ESA (European Space Agency) mission called SMOS (Soil Moisture and Ocean Salinity) aims to determine soil moisture over land and ocean salinity over oceans [5]. It has been steadily operating for over a decade since its launch on 2 November 2009 and is fully supported for enhanced validation [4–7]. SMOS's only payload MIRAS (Microwave Imaging Radiometer using Aperture Synthesis) is a passive microwave 2-D interferometric radiometer consisting of a central hub and three deployable arms [8–10]. Figure 1a shows the MIRAS configuration diagram [9]. The observed Tb (Brightness Temperature) can be modeled through a flat sea component and rough sea contribution [6]. Tb reflects sea surface emissivity $e$, which depends on SSS, SST (Sea Surface Temperature), frequency $f$, incidence angle $\theta$, and polarization $P$ [4]. $Tb = F(SSS, SST, f, \theta, P)$. So, knowing other parameters, SSS is retrieved. $SSS = F(Tb, SST, f, \theta, P)$.

The SMOS mission operates in a specific way. Firstly, it measures Tb in four polarization modes: HH, VV, HV, and VH. Among them, HH and VV modes set three arms along the same polarization direction H or V, while HV and VH modes set one arm in H/V and two others in V/H [11]. Secondly, MIRAS measures Tb over a wide range of incidence angles [8,9]. Therefore, every geographical point will be observed multiple times, producing a series of Tb values related to various incidence angles. Figure 1b shows SMOS's measurement mode observation geometry [9].

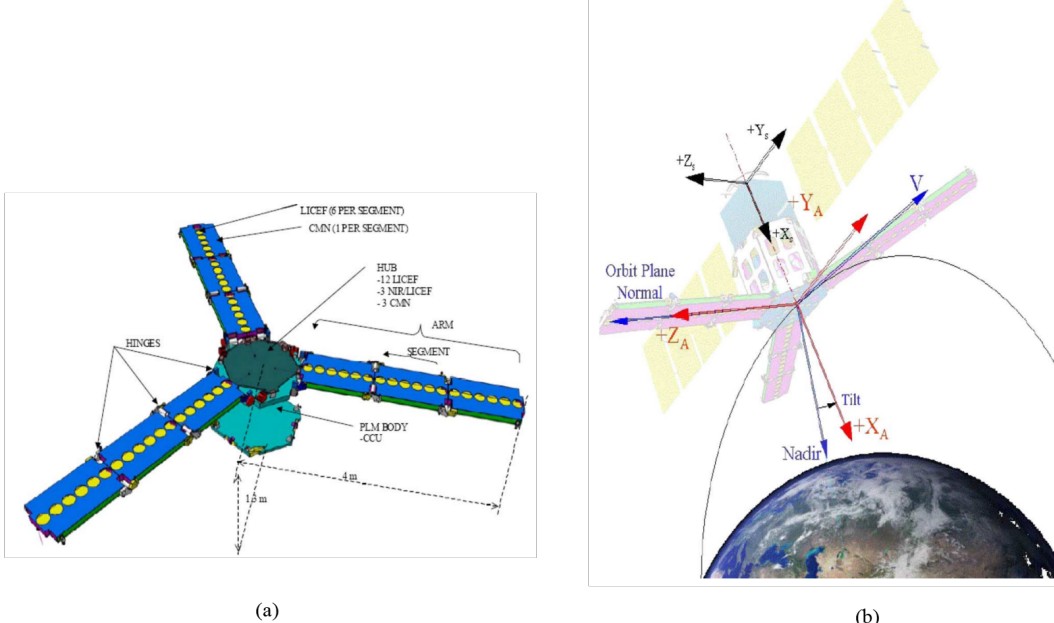

**Figure 1.** The payload MIRAS [9]. (**a**) Deployed MIRAS configuration diagram. (**b**) Measurement mode observation geometry.

SMOS operates in the EESS (Earth Exploration Satellite Service) passive band, 1400–1427 MHz, which is protected from all emissions according to provision No.5.340 of RR (Radio Regulations) of ITU-R (the International Telecommunication Union—Radiocommunications Sector) [12]. However, a large amount of RFI (Radio Frequency Interference) has been discovered worldwide and it is negatively impacting SMOS products [13]. Generally, RFI is caused by two types of sources [14,15]:

1. Unauthorized emissions within the protected passive band coming from active sources;
2. Unwanted emissions from active services operating in adjacent bands.

Considering RFI intensity and the unique layout of MIRAS, the contamination manifested in the field of view can be classified into two types:

1. Prominent contamination: usually caused by on-site RFI emissions;
2. Moderate contamination: comes from powerful RFI emissions on nearby land through the secondary lobes' tails [12].

To mitigate the impact, Anterrieu [16] measured RFI in the visibility domain, which is advantageous in detecting at a very early signal processing stage. Li [17] applied a spectral difference method for RFI measurement in the Aqua AMSR-E radiometer channels. Soldo [18] presented a method to obtain snapshot-wise information about RFI sources. Park [19] proposed detection based on the MUltiple SIgnal Classification (MUSIC) algorithm. However, no technique is able to eliminate RFI effects without it leading to data loss [14], and national authorities enforcing the ban on illegal transmitters is the only way to solve the problem at present.

To improve ocean salinity product quality, the SMOS team has set up a combined RFI detection strategy applying to various signal processing stages. Table 1 [12] lists a summary of these attempts.

**Table 1.** Combined RFI detection strategy by the SMOS team.

| Level | Based on |
| --- | --- |
| Level 1A | Temporal evolution of zero baseline |
| Level 1C | Observed intensity of RFI source from a known RFI list |
| Level 1C | Observed intensity of RFI source from a known RFI list |
| Level 2 | Min/max expected surface brightness temperature |
| Level 2 | Excessive spatial standard deviation in snapshot |
| Level 2 | Outlier detection |

After collaboration between the scientific community and international authorities for many years, a lot of RFI emissions have been removed successfully, leading to a considerable increase in the retrieved ocean salinity data. However, in terms of scientific significance, it is more valuable to quantify intensity or probability of the contamination and try to alleviate its influence proactively, instead of simply discarding corrupted data passively.

SMOS products over continents and oceans are split and processed separately. This study only focuses on RFI over oceans; the same idea can be applied to continents with specific alterations though. Over oceans, a few prominent contamination events could have come from radar devices deployed on ships. However, the main problem is from the secondary lobes of strong sources on land, which normally manifests as moderate contamination. This study proposes a comprehensive method to detect RFI of both prominent and moderate types. We deliberately put forward the conception of the "RFI map" to statistically present the state of RFI, which, as far as we know, is proposed for the first time in this context. The expected contribution of the "RFI map" is that it will be prepared as an auxiliary for SSS retrieval, thus assisting in ocean salinity acquisition in these extremely RFI-contaminated areas where the conventional methods have failed to give acceptable results.

## 2. Materials

It is well known that the stronger the RFI emission, the more widespread the contamination [12]. Therefore, even though 99% of RFI sources are located over land [14], some extreme irregular emissions can cause detrimental consequences on coastal waters. As a marginal sea in Southeast Asia, SCS (South China Sea) is one of the regions where the contamination from surrounding lands is so severe that masses of data have to be discarded during ocean salinity retrieval [20,21]. Under the circumstances, the reprocessed teams can only blank out some areas in their level3 and level4 SSS products. Figure 2 shows global sea surface salinity maps from CATDS (Centre Aval de Traitement des Donn Données SMOS) [22] and BEC (Barcelona Expert Center) [23]; both SSS maps present gaps in SCS region (red boxes). Therefore, this study chooses SCS and its surrounding area within the range of 4°N–25°N, 105°E–125°E in 2018 as the study object.

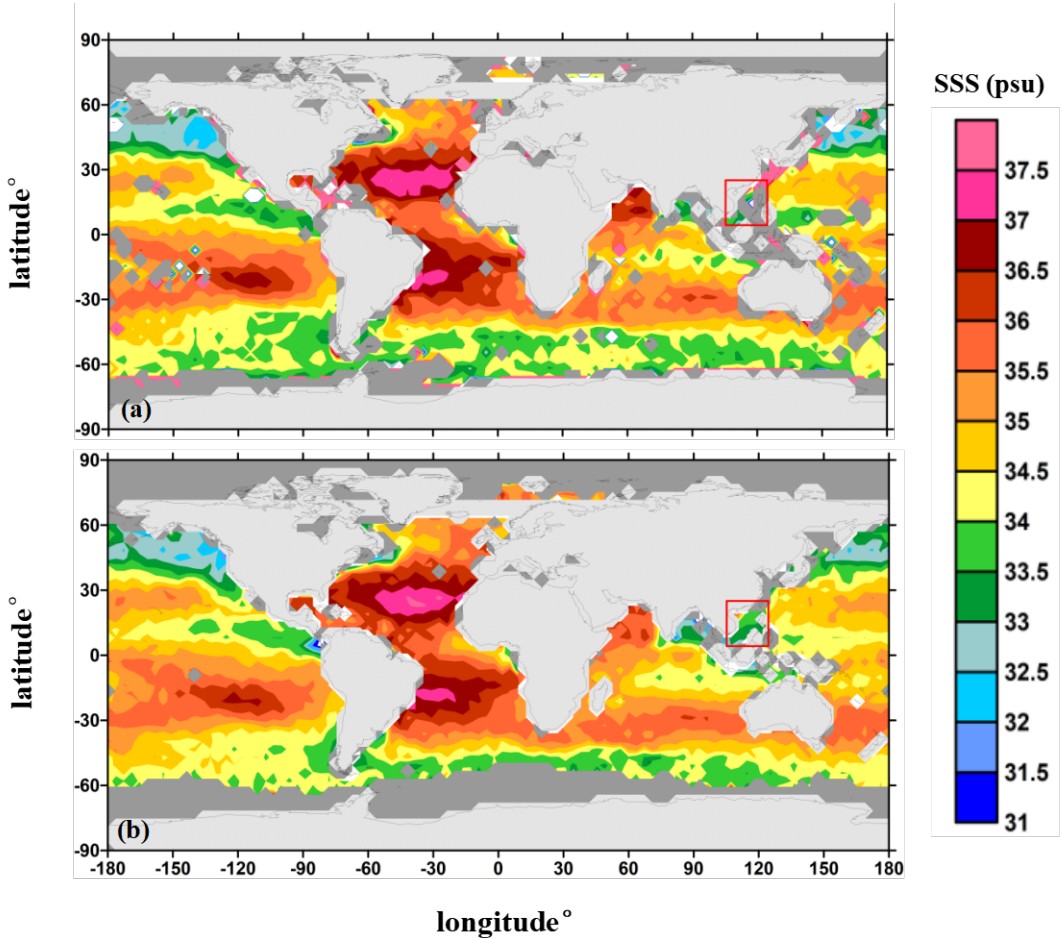

**Figure 2.** Level 3 SSS global products. Insufficient data for ocean salinity retrieval leaves gaps represented by dark gray; SSS global products provided by two reprocessed teams (**a**) CATDS—Centre Aval de Traitement des Données SMOS, France. (**b**) BEC—Barcelona Expert Center, Spain. SCS region is marked by the red box.

SMOS provides level1c Tb (Brightness Temperature) product and level2 OSUDP (Ocean Salinity User Data Product) openly through ESA SMOS online dissemination service. Whereas level2 OSDAP (Ocean Salinity Data Analysis Product) is restricted to only expert users, which can be accessible after sending a request for authorization. Table 2 gives a brief introduction of these products. It only shows the variables applied in this study; for more information, the reader can refer to [24,25]. These three products correspond one-to-one because the level2 products are produced based on the level1c products. However, as we mentioned earlier, SMOS observes each geographic point multiple times from different incidence angles. So, under the same Grid_Point_ID, variables related to Tb in level1c product and level2 OSDAP have many-to-one ocean salinity in level2 OSUDP.

**Table 2.** SMOS products.

| Products | Level 1c Tb | Level 2 OSUDP | Level 2 OSDAP |
|---|---|---|---|
| Description | Multi-incidence angle brightness temperatures, geo-located in an equal-area grid system. | The retrieved sea surface salinity, including theorical estimate of accuracy and flags for the product quality. | Quality control information in SSS retrieval for more advanced users. |
| [1] Contents | Grid_Point_ID<br>Latitude (deg)<br>Longitude (deg)<br>* BT_Value_Real (K)<br>* BT_Value_Imag (K)<br>* Pixel_Radiometric_Accuracy (K)<br>* Incidence_Angle (deg)<br>* Snapshot_ID | Grid_Point_ID<br>Latitude (deg)<br>Longitude (deg)<br>SSS_corr (psu)<br>SST (C) | Grid_Point_ID<br>Latitude (deg)<br>Longitude (deg)<br>* Diff_TB (K)<br>* Snapshot_ID |
| Type | preprocessed data | preprocessed data | preprocessed data |
| Format | Earth Explore (EE) file<br>NetCDF | Earth Explore (EE) file<br>NetCDF | Earth Explore (EE) file |

[1] Contents mentioned here are only what we used in the study, the other variables not listed are mostly the intermediate products in SSS retrieval, please refer to [24,25] for more information. * These variables are multiple under the same Grid_Point_ID, distinguished by Snapshot_ID.

## 3. Quantitative Measurement

At present, no single method is capable of filtering all RFI contamination [12]. To flag both prominent and moderate contamination levels, we propose RFI detecting and measuring in two aspects separately. Before that, because of the systematic error induced by SMOS hardware and software deficiency, an important step named OTT (Ocean Target Transformation) has to be done on level1c Tb. Figure 3 shows the overall pipeline of the proposed method.

### 3.1. Ocean Target Transformation

Systematic bias is observed between Tb measured by MIRAS and Tb used for ocean salinity retrieval model [26,27]. The mismatch may have many causes, such as imperfect instrument calibration or inadequate image reconstruction. The SMOS team has proposed the OTT (Ocean Target Transformation) technique to mitigate this error regardless of where the bias originates [11]. OTT generation has to be performed on a homogeneous ocean far from land. First, biases are determined in the xi-eta antenna frame (xi-eta coordinates in the SMOS field of view [25]) by averaging error differences between the forward model and the MIRAS measured TB values over a sequence of snapshots.

$$\text{OTT offset}^i = \text{MIRAS measured Tb}^i - \text{forward model Tb}^i \tag{1}$$

$$\overline{\text{OTT offset}} = \frac{1}{N} \sum_{i=1}^{N} \text{OTT offset}^i \quad i = 1, 2, 3, \ldots, N \tag{2}$$

Then, during L2OS (Level 2 Ocean Salinity Retrieval) processing, the offset will be applied to Tb used for ocean salinity retrieval.

$$\text{Tb used for retrieval} = \text{MIRAS measured Tb} - \overline{\text{OTT offset}} \tag{3}$$

The SMOS level2 OSDAP2 product has reserved every Tb difference according to its position on the xi-eta frame in MIRAS view. To ensure the accuracy of the produced RFI

map, we first alter level1c Tb with these differences; hence, all the subsequent discussions are based on the corrected Tb values.

$$\text{Tb used for RFI detection} = \text{level1c Tb} - \text{diff Tb} \tag{4}$$

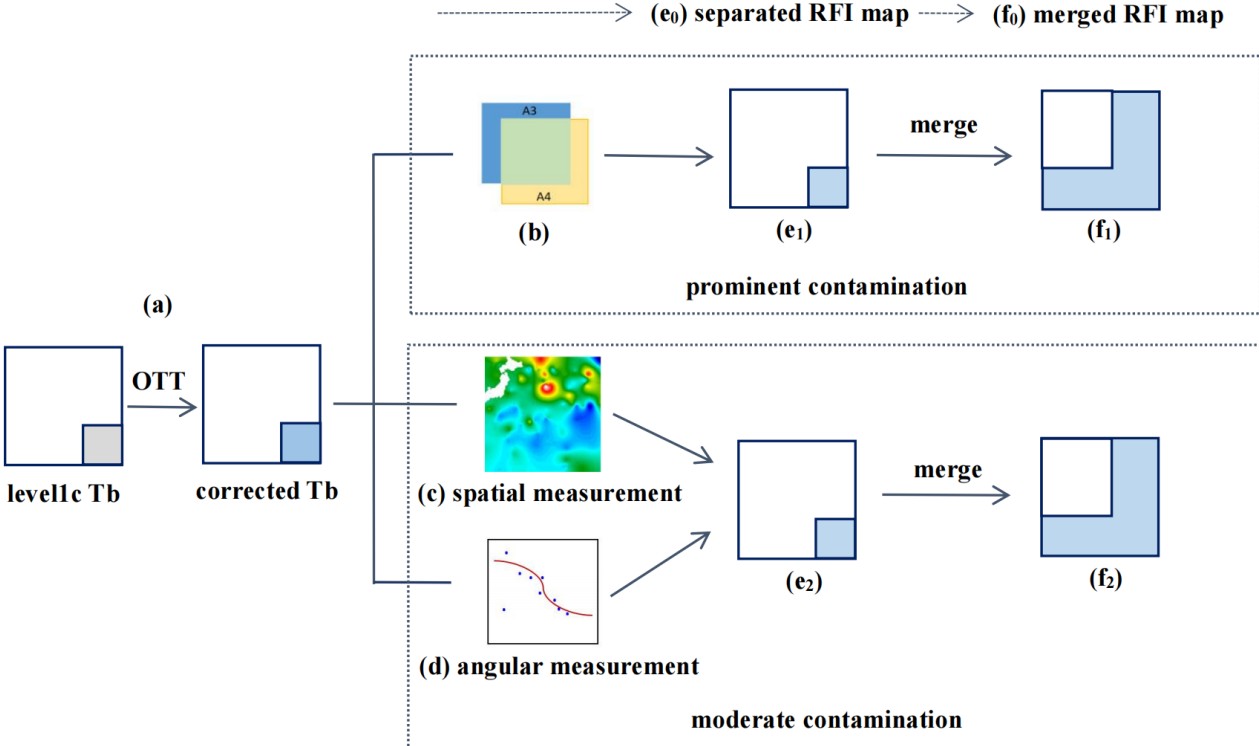

**Figure 3.** RFI detecting and measuring pipeline. Before the measurement, the level1c Tb must undergo (**a**) OTT to mitigate the instrument's systematic error. Then, the corrected Tb is used to detect the prominent and moderate contamination simultaneously. (**b**) The A3 and A4 Stokes parameters are utilized to locate the prominent contamination. (**c**) The spatial and (**d**) angular measurements are operated in parallel to present the moderate contamination. Each SMOS pass will produce two (**e0–e2**) separated RFI maps (prominent and moderate) reflecting the condition of only the specific area. Furthermore, (**f0–f2**) the merged RFI map will be presented after merging all separated maps of the area of interest over an extensive period.

*3.2. Prominent Contamination*

Prominent contamination caused by illegal on-site emissions, such as large cruisers or coastal ports, can have a devastating effect on local SSS acquisition. This study utilizes Stokes parameters to locate RFI targets. When MIRAS operates in full polarization mode, the measurement cycle is –HH–HV–VV–VH–. In the SMOS level1c product, both HV and VH modes feed the brightness temperature data field with real and imaginary parts. Under this operating mode, the four modified Stokes vectors are defined as (5) [11].

$$\begin{Bmatrix} A_1 \\ A_2 \\ A_3 \\ A_4 \end{Bmatrix} = \frac{\lambda^2}{KB\eta} \begin{Bmatrix} \langle |E_h|^2 \rangle \\ \langle |E_v|^2 \rangle \\ 2\text{Re}\langle E_v E_h^* \rangle \\ 2\text{Im}\langle E_v E_h^* \rangle \end{Bmatrix} = \begin{Bmatrix} T_{xx} \\ T_{yy} \\ 2\text{Re}(T_{xy}) \\ 2\text{Im}(T_{xy}) \end{Bmatrix} \tag{5}$$

where $\lambda$ is the radiometer's wavelength, K is the Boltzmann constant, B is the bandwidth, and $\eta$ is the medium impedance (air). $E_h$ and $E_v$ are the two orthogonal components of the plane wave.

In Equation (5), the third Stokes parameter $A_3$ is the difference between crossed linear polarization oriented at $45°$. Furthermore, the fourth Stokes parameter $A_4$ can be interpreted as the difference between left-hand and right-hand circularly polarized components [11]. Although both are expected to be negligible at L-band over natural objects, an RFI target is likely to contribute more to $A_3$ and $A_4$, since artificial sources generally do not align with the polarimetric axes of the sensor [28]. Therefore, using unusually large $A_3$ and $A_4$ to detect RFI is quite reasonable and has been confirmed by airborne campaigns [29]. In our prominent contamination measurement, a parameter $Q$ compounded by $A_3$ and $A_4$ is designated as (6). According to the above analysis, where $Q$ is abnormally larger than others, there is a high likelihood that the point has been RFI contaminated.

$$Q = \frac{\sqrt{A_3^2 + A_4^2}}{4} = \frac{\sqrt{2\text{Re}^2(T_{xy}) + 2\text{Im}^2(T_{xy})}}{4} = \sqrt{\text{Re}^2(T_{xy}) + \text{Im}^2(T_{xy})} \qquad (6)$$

*3.3. Moderate Contamination*

Moderate contamination is often from illegal emission sources on nearby land. Although the local impact may not be as severe as the on-site prominent contamination, its influence can reach the middle ocean and impair the data quality over a wide range. This study measures moderate contamination by spatial and angular approaches in parallel, with the final output being the average of the two.

3.3.1. Spatial Measurement

The principle of the spatial measurement is directly locating irregular Tb values. Brightness temperature Tb measures the radiance traveling from the target to the satellite. It is not the actual temperature of the object but the temperature of a blackbody that would emit the same amount of radiation as the target in the specified spectral band [30]. Under this definition, the relationship between the brightness temperature of sea surface and its actual temperature is formulated as (7).

$$\text{Tb} = \text{SST} \times e \qquad (7)$$

SST is the sea surface temperature, and $e$ is the surface emissivity at L band. According to the formula, when SST is given, the observed Tb should not exceed the expected value over a certain threshold [12,31]. Otherwise, the concerning area may suffer abnormal interference. The expected Tb for each geographic pixel can be expressed as (8).

$$\text{Tb}_{\text{model}} = (\text{SST} + 2\sigma_T) \times e_{\text{max}} + 2\text{PRA} \qquad (8)$$

$$\text{Tb} - \text{Tb}_{\text{model}} > 50 \qquad (9)$$

We use SST coming from ECMWF (European Center for Medium Weather Forecast). $\sigma_T$ is the forecast's uncertainty and is normally assigned to 2.5 k. The surface emissivity $e$ takes the maximum value of 1.0. PRA is the radiometric accuracy at each pixel, which could be obtained from SMOS level2 OSUDP. Margins of double $e$ and double PRA are added to improve the confidence level. The pretests suggest it is appropriate to appoint the threshold to 50 K; thus, pixels in which observed Tb exceeds the modeled Tb over 50 k will be considered RFI contaminated.

3.3.2. Angular Measurement

Among four SMOS polarization modes, the angular measurement only applies to HH and VV polarization. In these two modes, the level1c Tb is a real number without the imaginary part. We found that for the same geographic point contained in multiple SMOS snapshots, the observed Tb values typically appear as a smooth curve related to a series of incidence angles. Furthermore, this Tb-angle curve shows opposite behavior

under different polarization, that is, Tb in HH mode decreases with increasing incidence angle, while Tb in VV mode increases with it (Figure 4). Studies believe that this angular relationship of one point from a sequence of snapshots is much more deterministic than that of adjacent points from one snapshot in the spatial domain [32].

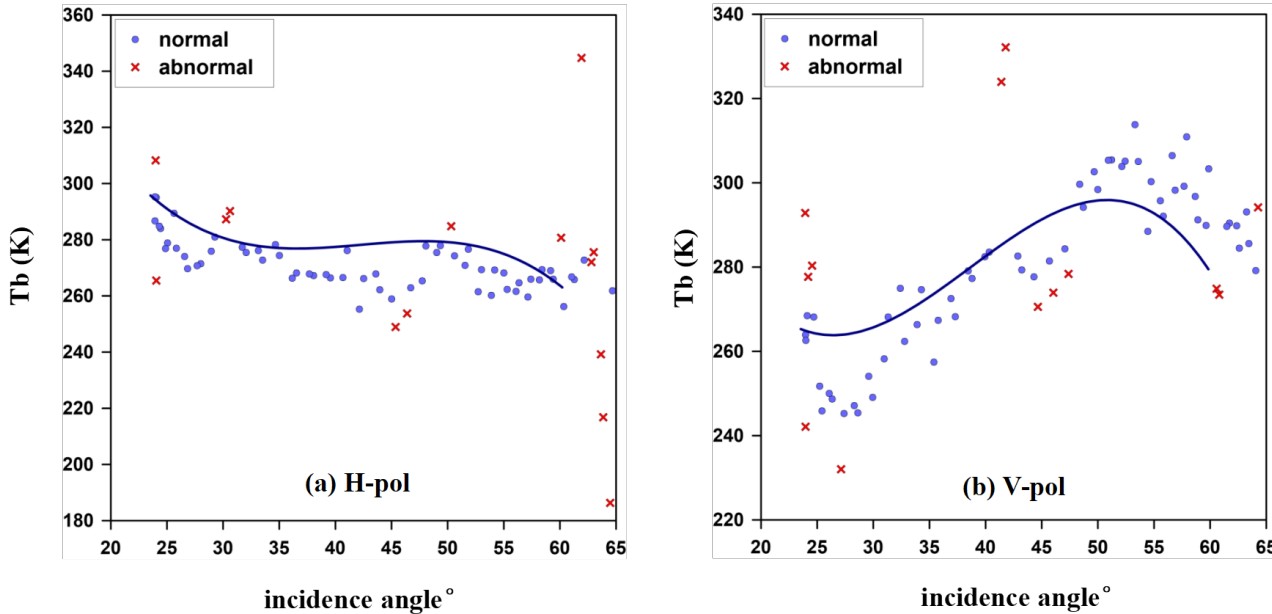

**Figure 4.** Angular measurement. (**a**) HH polarization; (**b**) VV polarization. The blue line is the third-order polynomial fit of all Tb values of one point from different snapshots, the blue spot is uncontaminated Tb, and the red cross is contaminated Tb, respectively.

Inspired by the idea, a point contained in multiple snapshots should exhibit irregular Tb behavior as a function of incidence angle, when RFI presents in that point. Based on the above principle, the angular measurement starts off with HH and VV mode separately. First, all snapshots containing the tested point are collected and initially filtered. When four or more snapshots are collected, we fit a third-order polynomial over the data using the least-squares method. We then compute the difference from each Tb to the cubic fit and make the threshold 1.5 times the average of the difference. Finally, the point farther from the threshold will be marked as RFI corrupted.

### 3.4. Merging

Two kinds of contamination, including the prominent and the moderate, are measured as described. In each of them, there are two-hierarchy RFI maps graded by the procedure. The primitive one is the separated RFI map, which is built on each SMOS pass. On top of that, the merged RFI map will accumulate counters of corresponding separated maps during the period of interest.

At the separated RFI map stage, as we discussed earlier, the prominent contamination measurement exploits Stokes parameters and calculates Q of every point. Furthermore, the moderate contamination measurement derives the percentage of abnormal Tb to all Tb values in the spatial and angular domain together.

While the separated RFI map can report the instant condition based on the contemporary SMOS pass, the merged RFI map shows the overall situation of a more extended period. After obtaining all separated RFI maps within the time interval of interest, the merging process will:

1.  Concatenate every piece of area these passes covered;
2.  Average RFI statistics of separated maps if points overlap.

Figure 5 is a simplified demonstration of the merging process. Firstly, a vector $\vec{V} = \{v_1, v_2, v_3, \ldots, v_n, \ldots, v_m\}$ is adopted, where $m$ is the number of geographic points in the target region, with element $v_n$ representing the RFI statistic of the point $n$. Above all the separated RFI maps related to the study region, we sum up every element $v_n$ separately, and divide them by the times of the corresponding point's occurrence. Note that some points in the separated maps may not be in the region of interest and need to be discarded. Finally, the output $v_n$ forms the merged RFI map of the target region.

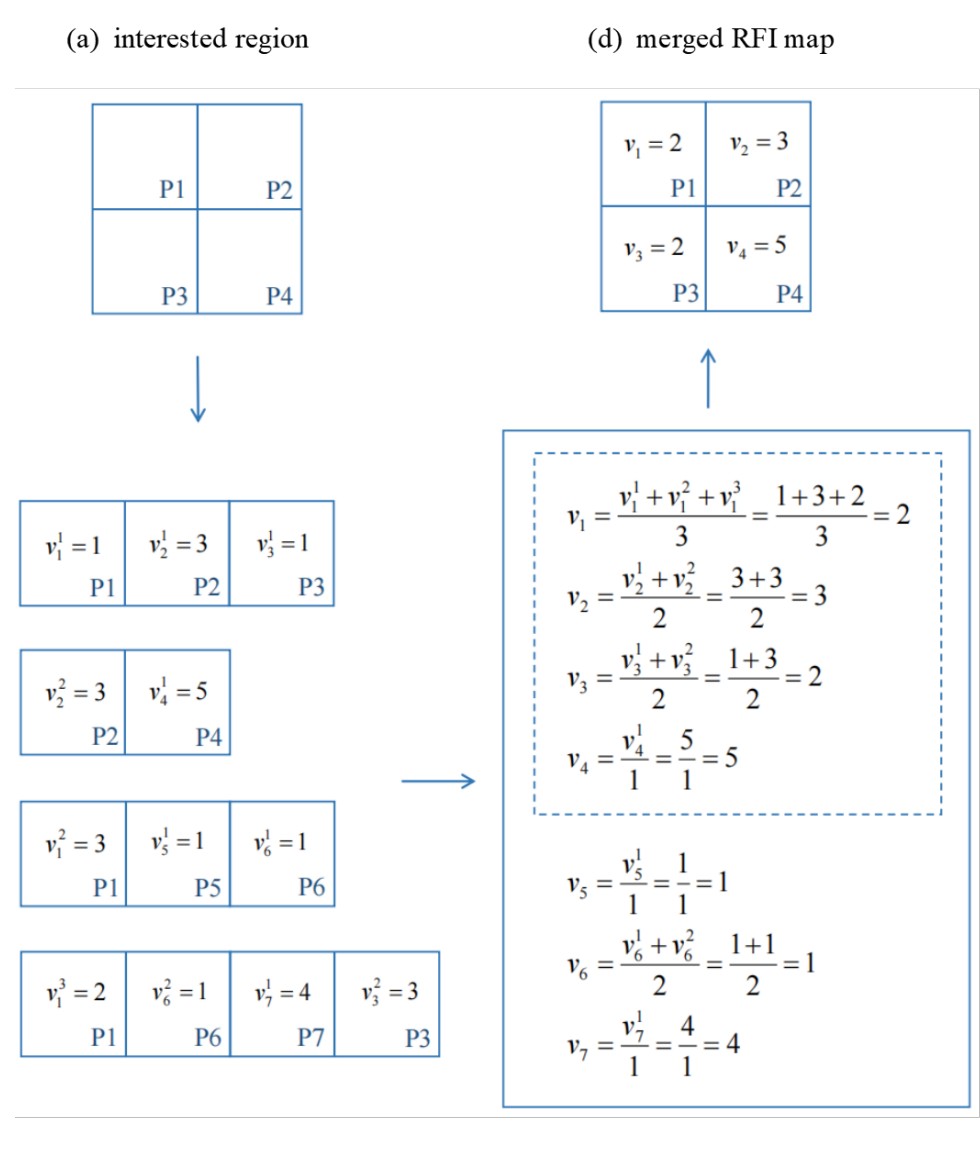

**Figure 5.** (**a**) Supposing there are four geographic points, P1, P2, P3, P4, in the region of interest. (**b**) The element $v_n$ represents the RFI statistic of the corresponding geographic point P$n$, and $v_n$ and P$n$ correspond one-to-one. (**c**) In the merging process, all the RFI statistics are summed up by a different element $v$ (or point P). For example, P1 exists in the first, the third, and the fourth separated maps, so the three $v_1$ are added together, and the process is the same for other elements. Then, the sum is divided by the point's times of occurrence, so the denominator is 3 in the case of $v_1$. (**d**) In this demonstration, $v_5$, $v_6$, and $v_7$ are not in the area of interest and will be dropped in the final merged RFI map.

## 4. Results and Discussion

### 4.1. The Separated RFI Map

The proposed method produces two kinds of RFI maps, with a different emphasis on its own. Relatively, The prominent contamination is locally severe, while the moderate contamination is milder but more widespread over the ocean. Both the prominent and the moderate separated RFI maps are produced when the method applies to one individual satellite pass. Figure 6 displays the separated RFI maps of three SMOS passes taken at different times on 15 January 2018. Due to the limited satellite swath width, each picture can only exhibit part of the SCS region (Figure 6c,f,i). The results show that the western SCS suffered notable prominent contamination on this day (Figure 6g), most likely from the heavy traffic along the coasts of Vietnam and Cambodia. In the case of the moderate contamination, the northern SCS is slightly more severe than the southern part (Figure 6b), indicating the negative influence from China, especially from big cities such as Hong Kong, Guangdong, and Taiwan province. The most significant advantage of the separated RFI maps is that they can display instant conditions of the shooting time, making it possible for researchers and authorities to take action immediately.

### 4.2. The 9-Day Merged RFI Map

While the separated RFI map can only exhibit a particular area involved with a specified pass, the merged RFI map will give more information for a larger region and a more extended time. In general, the merged RFI map can be comprehended in two aspects:

- Intensity: it represents the average of RFI intensity for a given period;
- Probability: it gives an intuition of how likely RFI is to exist during that time.

Figure 7 displays the merged RFI maps based upon the separated RFI maps on nine successive days in June, September, and December, relatively. The time period of nine days is chosen because the CATDS CEC-LOCEAN debiased SMOS level3 product contains a global sea surface salinity map of nine days [22], which we can utilize to compare the error with the merged results. Figure 7a,c,e present the prominent contamination, which points out the accurate location of RFI sources in the ocean. The results make it apparent that most areas on wide SCS are free from illegal emission, except for several parts in the west distributed over the coasts of Southeast Asia, such as in the Beibu Gulf and those near Vietnam and Cambodia. These emissions may come from big harbors. For instance, Fangcheng harbor in the Beibu Gulf, one of China's most important seaports, is very likely a major RFI emission source. Meanwhile, as discussed above, fishing and transporting vessels along the coasts in the western SCS are another common cause of the prominent contamination. In comparison, the positions of RFI focus in June, September, and December were not completely overlapping, meaning that the RFI emission sources are not constant and always change with the nearby cities' regulations and trades. For example, the prominent contamination in December (Figure 7e) reduced remarkably, possibly due to cold weather cutting back marine business.

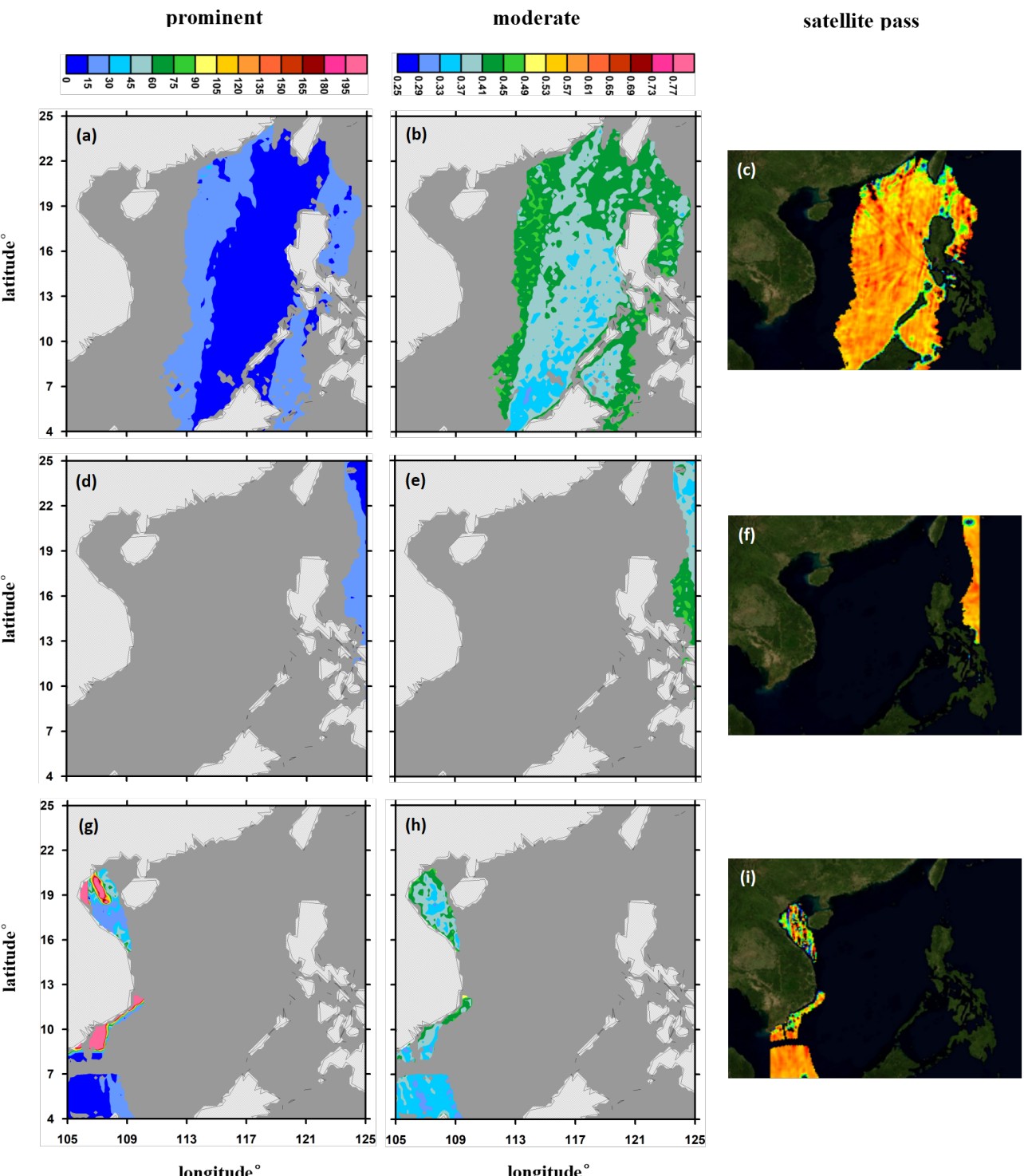

**Figure 6.** The separated RFI map. The prominent RFI maps (**a**,**d**,**g**) and the moderate RFI maps (**b**,**e**,**h**) are built on three SMOS passes on 15 January 2018. (**c**,**f**,**i**) are corrected SSS of these passes provided by ESA SMOS team, (**c**) corresponded to (**a**,**b**) which was taken on 15 January 2018 from 09:53:49 to 10:47:04, (**f**) corresponded to (**d**,**e**) which was taken on 15 January 2018 from 20:44:14 to 21:37:33, and (**i**) corresponded to (**g**,**h**) which was taken on 15 January 2018 from 22:24:19 to 23:17:37. All images have been truncated into SCS region. The light gray in the images represents the land area, and the dark gray is the area that was not covered during the discussed satellite pass.

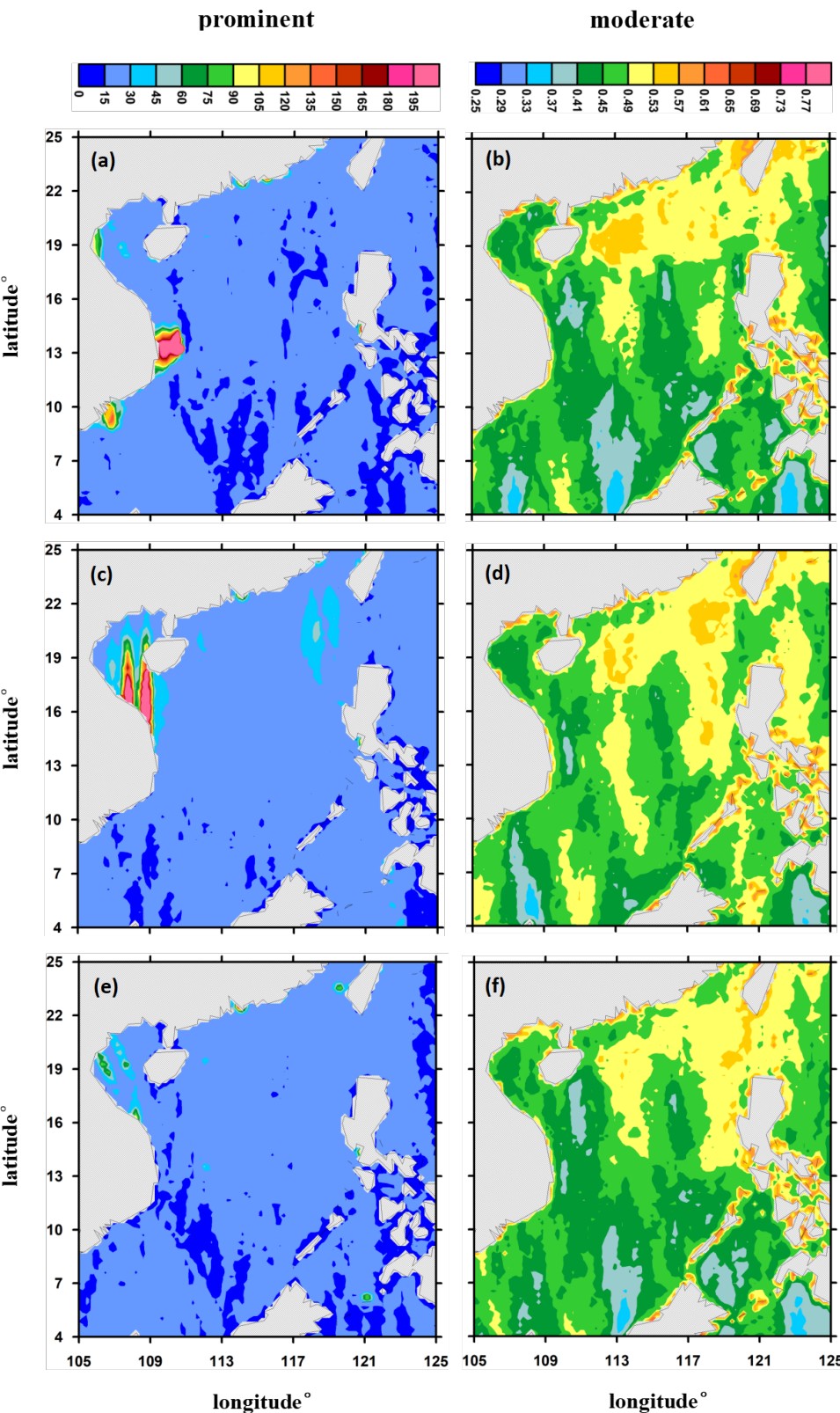

**Figure 7.** The 9-day merged RFI map. The prominent RFI maps (**a**,**c**,**e**) and the moderate RFI maps (**b**,**d**,**f**) are merged RFI maps of 9 days, presenting contamination in three different periods. Among them, (**a**,**b**) are RFI condition during 11 to 19 June 2018, (**c**,**d**) are for the time from 11 to 19 September 2018, and (**e**,**f**) are for the time from 12 to 20 December 2018. All images have been truncated to SCS region. The light gray in the images represents the land area.

As for the moderate contamination (Figure 7b,d,f), it seems that the whole SCS region has been interfered with to varying degrees, which is sternly warning us of the grim situation. In general, RFI in the northern part of SCS near China is much more severe, especially in the Luzon Strait, Taiwan Strait, and areas close to Hong Kong, Guangdong province, and Taiwan province. Other than that, waters around the Philippines and surrounding islands are also hard hit. The most possible explanation may be its proximity to the mainland; thus, it is largely influenced by these populous cities. Three moderate RFI maps of different periods look similar on the whole, yet some deviations are present in certain areas. For example, RFI contamination from 2018-09-11 to 2018-09-19 (Figure 7d) has extended to the southern part of SCS while the other two have only reached the middle part of SCS.

To validate the produced RFI maps, Figure 8 displays the absolute error between CATDS CEC-LOCEAN debiased SMOS level3 product [22] and the referenced CMEMS (Copernicus Marine Environment Monitoring Service) GOER (Global Ocean Ensemble Re-analysis) SSS product [33]. GOER was produced with a numerical ocean model constrained with data assimilation of satellite and in situ observations, and it was built to be as close as possible to the observations and in agreement with the model physics [33]. In this analysis, GOER products in the same time periods as in Figure 7 are collected and geographically matched to CATDS ocean salinity products, with the spatial window of 0.25°. Then, the absolute error is calculated with $\epsilon = SSS_{CATDS} - SSS_{GOER}$. As mentioned earlier, CATDS SMOS level3 product has already dealt with RFI and land-sea contamination, thus leaving considerable areas blanked out. Figure 8 shows that Luzon Strait and the Philippines' areas are blanked out for the extreme contamination. In addition, the entire Beibu Gulf is blanked out as a major emitter, as presented here (Figure 7a,c,e). Comparing the three images in Figure 8, Figure 8a,c suffer more in the northern SCS while the RFI of Figure 8b already extends to the middle and southern SCS, which is generally consistent with our produced RFI maps (Figure 7b,d,f).

### 4.3. The Yearly Merged RFI Map

Figure 9 displays the measured prominent and moderate RFI maps for the entire year of 2018. The results show that most parts are free from prominent contamination (Figure 9a), except for a few points on the west SCS near Vietnam. The moderate RFI map has a layered structure on the whole, in which higher latitude means more severe contamination, in accordance with the trend illustrated in Figure 7b,d,f.

According to the definition and characteristics of two kinds of contamination, the prominent contamination implies RFI sources on the spot such as trading ports and trans-porting vessels, and the moderate contamination comes from illegal emissions on the mainland due to urban activity. Therefore, combined with the actual places of origin, coastal ports in Vietnam and human activities in Hong Kong, Guangdong, and Taiwan are the primary culprits of RFI in the SCS region.

### 4.4. Generalization

It should be stressed explicitly that the proposed RFI quantification method is not limited to the study area. Theoretically, it applies to any water region in the world. To testify the generalization of the proposed method to other regions and also to grasp the RFI condition in the open ocean far from land, we select the tested area in the central Pacific within the range of 4°N–25°N, 142°W–122°W and compute its merged RFI maps during 11–19 June 2018. From the results (Figure 10), we see that both prominent and moderate contamination is extremely low in the tested area, indicating very few or no RFI presented in this unattended region.

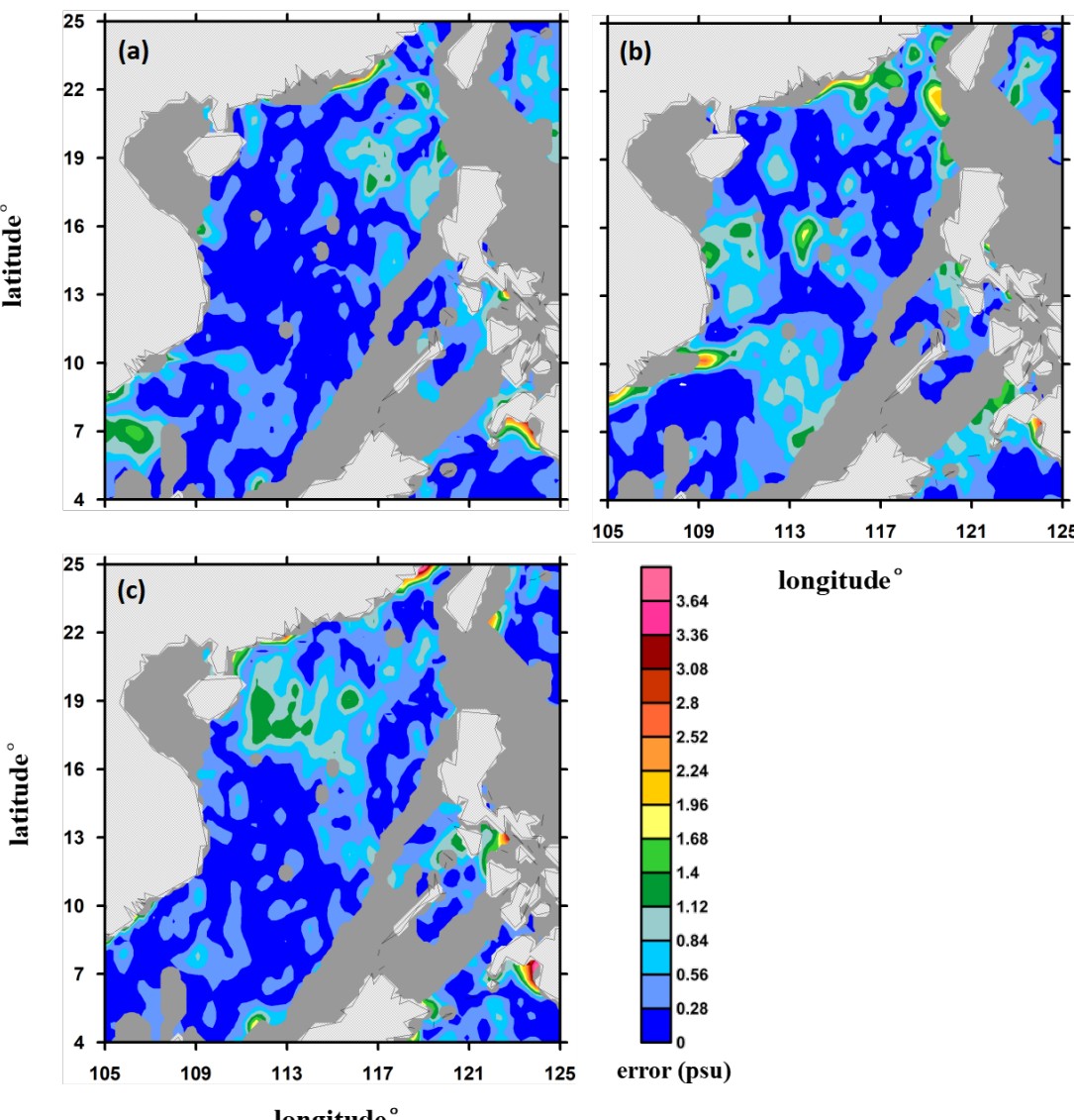

**Figure 8.** The absolute error of CATDS SMOS level3 product. CMEMS GOER sea surface salinity product is chosen as the reference for comparison. (**a**) is the error between the CATDS SMOS level3 product and the CMEMS referenced SSS from 11 to 19 June 2018, (**b**) is the error from 11 to 19 September 2018, and (**c**) is the error from 12 to 20 December 2018. All images have been truncated to the SCS region. The light gray in the images represents the land area and the dark gray is the area that is blanked out because of large error induced by RFI and other factors.

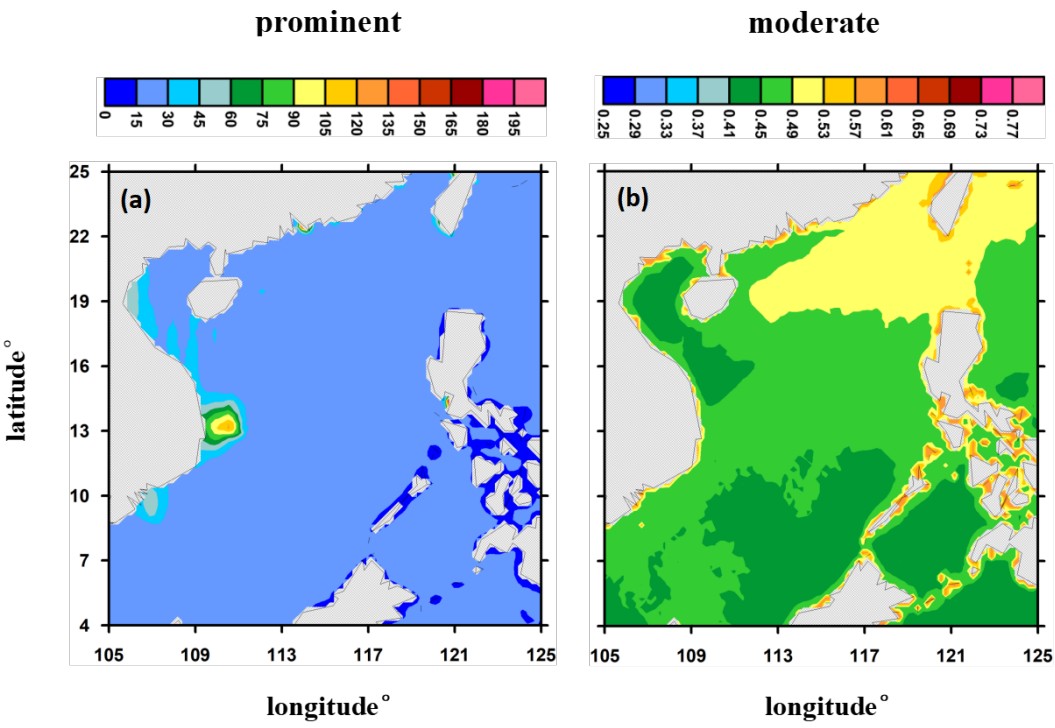

**Figure 9.** The yearly merged RFI map. (**a**) The prominent RFI map and (**b**) the moderate RFI map are yearly merged RFI maps of 2018, presenting contamination condition for the whole year. All images have been truncated to SCS region. The light gray in the images represents the land area.

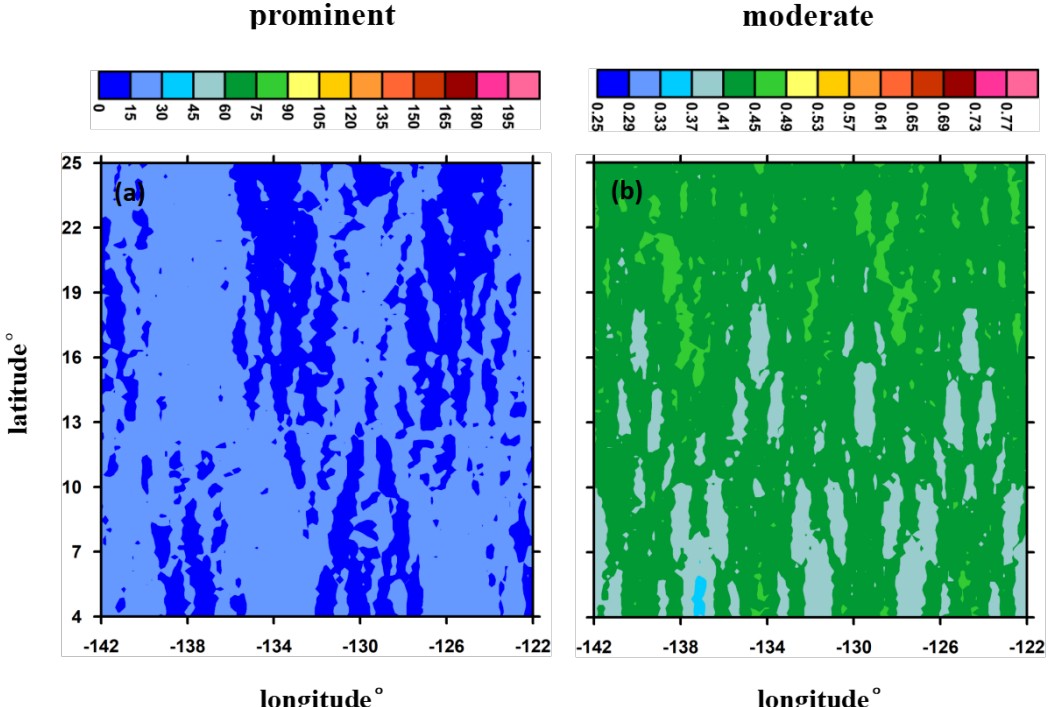

**Figure 10.** The RFI map of the area in the central Pacific. (**a**) The prominent RFI map and (**b**) the moderate RFI map are merged RFI maps of the tested area in the central Pacific within the range of 4°N–25°N, 142°W–122°W from 11 to 19 June 2018. All images have been truncated within the tested area.

Moreover, both prominent and moderate RFI maps express an alternating feature, which we consider to imply the satellite trajectory. According to the satellite plan [8,9], SMOS observations of high incidence angles are much more sparse than those of low incidence angles; therefore, the collected Tb values from the periphery are far lower than from the center of each pass. The significant difference in Tb numbers leaves satellite footprints on the resulting RFI map that, although covered by heavy contamination in SCS, appeared in the unattended open ocean. Figure 11 is the BEC-provided SSS uncertainty [23] of the tested region. It has a similar alternating pattern to the feature manifested in Figure 10 and therefore verifies our assumption.

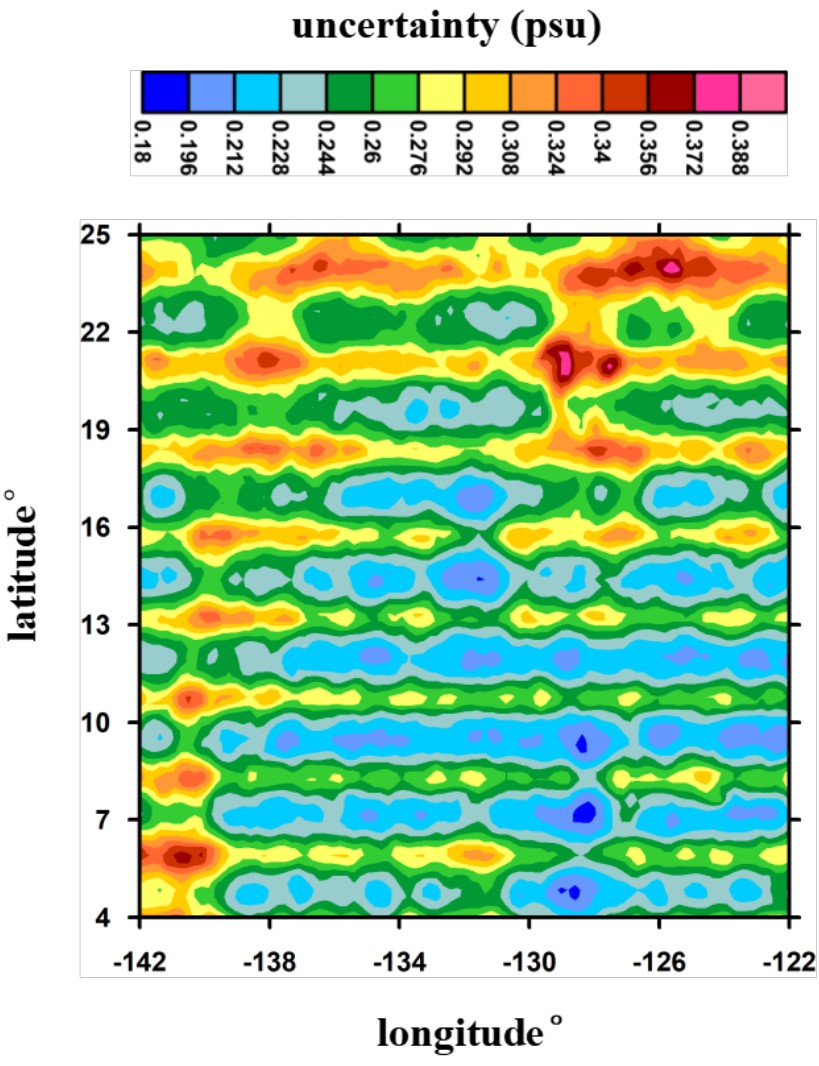

**Figure 11.** Sea surface salinity uncertainty of the tested region. Provided by BEC—Barcelona Expert Center, Spain. The image has been truncated into the tested region).

## 5. Conclusions

It is essential to protect the passive L band 1400–1427 MHz from other illegal emissions that disturb earth observation by SMOS and other satellite missions. Unfortunately, despite many actions taken by international authorities, radio frequency interference is still significantly impacting ocean salinity retrieval around the world, especially in coastal waters near continents, where lots of data have to be discarded in the final sea surface salinity product. The problem remains unless a method of quantifying RFI is found, which will be an auxiliary that assists in ocean salinity retrieval. For this purpose, the manuscript puts forward the "RFI map" for the first time in this context to help researchers comprehend RFI manifestation more intuitively and statistically.

The proposed RFI measuring method is able to detect both the prominent contamination, which is aroused by illegal emissions on-site, and the moderate contamination, which is usually from nearby continents. Two levels of hierarchical RFI maps are produced. The primitive one is the separated RFI map, built on a single SMOS pass, giving instant reports to relevant departments. The higher-level result is the merged RFI map, which will reveal the overall contamination condition of the region of interest for an extended period.

In this study, nine-day merged RFI maps of the SCS region are discussed and validated by comparing them with the CATDS SMOS level3 product error. In addition, the yearly merged RFI maps are presented and thoroughly analyzed, illustrating the overall status of 2018. In general, all the RFI maps convey the information of more severe contamination in the northern and western parts of SCS near the mainland, which hopefully will raise the alarm to the public.

The statistic "RFI map" is not only to flag whether the contamination exists but to compute how intense or how likely it is. Its ultimate goal is to prepare an auxiliary parameter in further sea surface salinity retrieval; therefore, it will help fill the gap of SMOS SSS products in highly contaminated areas, which are currently suffering unsolvable data loss.

**Author Contributions:** Conceptualization, M.X. and H.L.; methodology, M.X.; software, M.X.; validation, H.C. and X.Y.; formal analysis, M.X. and H.C.; investigation, M.X.; resources, M.X.; data curation, M.X.; writing—original draft preparation, M.X.; writing—review and editing, M.X. and H.L.; visualization, M.X.; supervision, H.L. and X.Y.; project administration, H.L. and X.Y.; funding acquisition, H.L. All authors have read and agreed to the published version of the manuscript.

**Funding:** This research was funded by the National Program on Global Change and Air-Sea Interaction (Phase II)—Parameterization assessment for interactions of the ocean dynamic system.

**Institutional Review Board Statement:** Not applicable.

**Informed Consent Statement:** Not applicable.

**Data Availability Statement:** Not applicable.

**Acknowledgments:** The authors would like to thank Piesat Information Technology Co., Ltd for their valuable advice. And we very much thank Manuel Martin-Neira and Kevin Mcmullan for providing images of Figure 1 in this manuscript. We also gratefully acknowledge the SMOS team and European Space Agency, Centre Aval de Traitement des Donn Données SMOS, Barcelona Expert Center, Copernicus Marine Environment Monitoring Service for providing materials and data in this article.

**Conflicts of Interest:** The authors declare no conflict of interest. The funders had no role in the design of the study; in the collection, analyses, or interpretation of data; in the writing of the manuscript; or in the decision to publish the results.

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
