# Peer review of "Quantitative Measurement of Radio Frequency Interference for SMOS Mission"

_remotesensing, doi:10.3390/rs14071669_

Round 1

Reviewer 1 Report

The authors propose a radio-frequency interference (RFI) measurement technique to measure and classify moderate to severe interference in coastal and oceanic regions. Using this technique, the authors map the severity of this contamination in the South China Sea and central Pacific Ocean. These results are targeted at ocean salinity measurement missions, wherein this information can be used to estimate the presence and severity of RFI, which affects the precision and reliability of the measured results.

[28] A simple diagram of the payload would significantly clarify the explanation in this paragraph.

[53] The formatting of Table 1 makes it difficult to read. Please consider using a left justification and adding some space between each entry to separate the block of text.

[83] Can you clarify what is contained in the data provided by these products? Is this raw or preprocessed data? What is the format? What are the contents?

[84] The color bar in Figure 1 needs a unit label.

[91] I would like to see a simple explanation of the processes depicted in Figure 2 inside the caption.

[92] The first sentence in this paragraph isn't clear and needs to be rewritten.

[97] Can you clarify what is meant by "xi / eta antenna frame"? I don't see these variables defined or used in the subsequent equations. Is this shorthand notation specific to this field or this mission?

[110] I believe "Stokes parameters" should have a capital S here and throughout the text.

[115] At what altitude does this mission operate? Is using the medium impedance of air a reasonable simplification in this context?

[125] From where does the definition of parameter Q defined in Equation 6 originate? Is this an original metric or a result from a different study? Can you briefly explain why it takes this form and how you expect this affect your data? I don't see any reference to this quantity in any of the remaining equations.

[139] What are the units of Tb? 

[161] Figure 3 is missing units on all of the labels. Please also consider using different markers for the "normal" and "abnormal" classifiers to help distinguish the two data sets.

[165] These 2 sentences are unclear and need to be restructured. Please consider something like "When 4 or more snapshots are collected, we fit a third-order polynomial over the data using the least-squares method."

[191] The explanation in this paragraph would be much stronger if it were supported by a diagram (like Figure 2).

[192] Overall, this section suffered from a lack of clarity, due to lack of (or unclear) supporting text, poorly structured sentences, equations with words rather than variables, and grammatical errors. After reading it again I was able to make sense of the explanation, but as it is written now it is difficult to follow. This section should be expanded to include the missing supporting information and carefully rewritten to improve clarity and delivery of said information.

[209] The color bars in Figure 4 need unit labels.

[232] You specifically mention "intensity" and "probability", but it isn't clear which you are representing in Figure 5, if either. This color bar is also missing units.

[243] Text is running into the margin on this line.

[257] The color bar in Figure 6 needs a unit label. It is unclear how you are defining "absolute error" if you are just referencing a different measurement technique. 

[268] The color bar on Figure 7 needs a unit label. 

[285] Can you explain what the SSS uncertainty is and how it was measured?

Reviewer 2 Report

The authors are required to explain the mechanisms of SMOS for retrieving sea surface salinity and the comparison with optical data such as MODIS.

The literature is not adequate as there are some references are required to be added to the introduction. I suggest the following reference:

Marghany, Maged, Mazlan Hashim, and Arthur P. Cracknell. "Modelling sea surface salinity from MODIS Satellite data." International Conference on Computational Science and Its Applications. Springer, Berlin, Heidelberg, 2010.

What is the importance of the salinity in coastal waters must be addressed?

There is an absence of the in situ measurements and ancillary data can be used for accuracy validation so how the authors can solve this matter?

What is the relationship between equation 8 and previous equations from 1 to 7?

The authors have jumped to conclusions without logical discussion.

At this stage, I cannot recommend the paper for publication.

Reviewer 3 Report

 Authors propose " Quantitative Measurement of Radio Frequency Interference for SMOS Mission". The topic is very interesting and the paper is, in general, well-written. I, however, have the following comments:

  1.   Authors need to re-write the Abstract in a more meaningful way
    example (Problem definition=> How existing methods are lacking
    => proposed solution and why it is better)

      2.   Related work section 1 is too short.  In the related work section authors are missing some important parts which are related to SMOS and paper related to Sea Surface Salinity. The author should discuss the following papers:

Guimbard, S., Reul, N., Sabia, R., Herlédan, S., Khoury Hanna, Z.E., Piollé, J.F., Paul, F., Lee, T., Schanze, J.J., Bingham, F.M. and Le Vine, D., 2021. The salinity pilot-mission exploitation platform (Pi-mep): A hub for validation and exploitation of satellite sea surface salinity data. Remote Sensing13(22), p.4600.

Akter, R., Asik, T.Z., Sakib, M., Akter, M., Sakib, M.N., Al Azad, A.S.M., Maruf, M., Haque, A. and Rahman, M., 2019. The dominant climate change event for salinity intrusion in the GBM Delta. Climate7(5), p.69.

3. In the figure 1 authors miss to put the unit of SSS

4.  How did the authors deal with unbalanced dataset 

Round 2

Reviewer 1 Report

- The rewrite of the abstract more clearly states what the authors are proposing and how it will be useful.
- The addition of the first two paragraphs significantly improves the context of the introduction and sufficiently introduces the SMOS mission to those unfamiliar with it.
- The addition of Figure 1 also improves the clarity of the introduction.
- The updated formatting of Table 1 is significantly easier to read.
- The addition of Table 2 sufficiently addresses my question about the contents of the SMOS products.
- The updated color bars are much easier to read and understand.
- The explanation added to Figure 3 is exactly what I asked for and significantly improves the clarity of the diagram.
- All suggested rewrites appear to have been resolved.
- The xi-eta footnote was helpful in explaining the terminology used by the SMOS mission.
- The derivations in Section 3 have been sufficiently expanded and clarified to address the previously mentioned issues.
- The addition of Figure 5 is a massive improvement to the clarity of this discussion.
- Formatting issues appear to be resolved.
- The figure labels have either been changed or explained and are now satisfactory.
- Thank you for the clarification on SSS uncertainty.

All of my previous comments have been sufficiently addressed, and I believe the manuscript is now suitable for publication. Thank you for your responses and your patience while we prepared a second reound of reviews.

Reviewer 2 Report

I think the authors have done excellent revision.